# Informed choice of modern Contraceptive Methods and determinant factors among reproductive age women in Eastern Africa countries: A multilevel analysis of demographic and health survey

Gebreeyesus Abera Zeleke[1]*, Birtukan Atena Negash[1], Kassaw Belay Shiferaw[2], Berhan Tekeba[3], Tadesse Tarik Tamir[3], Workie Zemene Worku[4], Alebachew Ferede Zegeye[5]

1 Department of Surgical Nursing, School of Nursing, College of Medicine and Health Science, University of Gondar, Gondar, Ethiopia, 2 Department of physiotherapy, School of medicine College of Medicine and Health Sciences University of Gondar, Gondar, Ethiopia, 3 Department of Pediatrics and Child Health Nursing, School of Nursing, College of Medicine and Health Sciences, University of Gondar, Gondar Ethiopia, 4 Department of community nursing, School of Nursing, College of Medicine and Health Science, University of Gondar, Gondar, Ethiopia, 5 Department of Medical Nursing, School of Nursing, College of Medicine and Health Sciences University of Gondar, Gondar, Ethiopia

* gebreeyesusabe143@gmail.com

## Abstract

### Background

Unwanted pregnancies arise from the discontinuation of many contraceptive methods or the failure to use current contraceptive services; this has been a public health concern in Sub-Saharan Africa, particularly in the eastern African countries. Informed choice of modern contraceptive method is an important indicator of family planning quality services. Evidence shows that informed choice of contraceptive methods lowers the potential risk of family planning discontinuation rate, misunderstanding of contraceptive method and unintended pregnancies finally lead to induced abortions. Therefore, the aim of this study was to ascertain the magnitude of informed choice of modern contraceptive methods user and its determinant factors among reproductive age women who are currently using modern contraceptive in Easter African countries.

### Methods

Secondary data analysis was conducted using data from the DHS eight Eastern Africa nations between 2012 and 2020. The total weighted sample was 6154 reproductive age women who used were modern contraceptive method. Stata version 14 was used to analysis secondary data. Determinants of informed choice were determined by using multilevel mixed-effects logistic regression model. Significant factors

**Data availability statement:** All relevant data are within the manuscript and its Supporting Information files.

**Funding:** The author(s) received no specific funding for this work.

**Competing interests:** Authors Declared that, there is no competing interest.

related with informed choice in multilevel mixed effect logistic regression model were decided when the p value of <0.05. The adjusted odds ratio (AOR) and confidence interval (CI) were used to interpret the outcome.

## Result

In East Africa, the magnitude of informed choice found to be 20.70. determinant factors such maternal age (20–35) years (AOR = 2.02 CI: 1.39, 2.93), (36–49) years (AOR = 2.44, 95% CI: 1.66, 3.61), attending secondary & higher education (AOR = 1.37 95% CI 1.02, 1.84), Media exposure (AOR = 1.25 95% CI 1.05, 1.49), visit health facility within 12 months were (AOR = 1.31 95% CI: 1.11, 1.54) were significantly associated with an informed choice among reproductive age women modern contraceptive method user.

## Conclusions and recommendation

This study concluded that only 20.7% reproductive age women using modern contraceptive method were an informed choice in Eastern Africa. The following factors were strongly linked to informed choice: maternal age, attending secondary and higher education, media exposure, and visiting a health facility within a year. Therefore, policies and initiatives targeting informed choice modern contraceptive methods and above critical determinants among reproductive age women in Eastern Africa (ages 15–49) will be designed by the government, ministry of health, and other relevant parties.

## Introduction

Family planning access has remained a fundamental component of services for sexual and reproductive health, according to in Cairo declaration, International Conference on population Development(1). The term "family planning" refers to all the services, including contraception, that assist individuals in preventing unintended pregnancies, deciding whether to have children, and deciding when to do so [1]. According to an analysis of data from 34 developing nations' Demographic Health Surveys (DHS), 38% of women lacked access needs for modern FP methods [2]. Another DHS analysis conducted in 15 different countries, 7–27% of former contraceptive users quit using it because of issues with the quality of care they received [3]. Only 30% of women said they received adequate information during counseling, according to Ethiopia's national performance monitoring action survey conducted between 2014 and 2018, suggesting that family planning counseling is typically of low quality [4]. In low- and middle-income nations, 74 million women experienced unintended pregnancies each year, resulting in around 25 million unsafe abortions and 47,000 maternal death annually [5]. Global estimates between 2010 and 2014 indicate that 45% of all induced abortions worldwide are unsafe [6]. According to a study done across five East African nations, around one in three couples will probably

stop using contraceptives after a year. Women most frequently cite desiring to have another child and actual or imagined negative effects as the two main reasons for stopping the usage of contraception [7]. People fundamentally have the right to obtain contraceptive information and services, and high-quality contraceptive information and services support people's autonomy to decide how many children to have and how spaced apart to have them, as well as offering a number of potential benefits, such as better mother and child health [8]. The freedom of a person or couple to make a freely chosen decision based on possibilities that they have a thorough understanding of, maybe through comprehension knowledge, is known as informed choice [9]. Unwanted pregnancies result from many contraceptive users quitting their methods or failing to utilize them, which is a public health concern in Sub-Saharan African nations, particularly those in Eastern Africa. [10]. A key indicator of high-quality family planning services is the ability to make an informed decision about a contemporary contraceptive method.

[11,12]. Evidence shows that informed choice of contraceptive method lowers the potential risk of family planning discontinuation rate, dissatisfaction of contraceptive method, unintended pregnancies which finally lead to induced abortions [13,14]. Informed choice has been crucial in assisting people in choosing a method that best suits their requirements and expectations in order to encourage the efficient use of contraceptives [15]. There is very little evidence that informed choosing of modern contraceptive methods is commonplace in various parts of the world. A cross-sectional study conducted Bangladesh 22% [16], Kenya 55.5% [17], Ethiopian (EDHS 2016) (36.2%) [18], in sub-Saharan Africa 49.47% [10]. To ensure the inclusion of varied demographic groups, for example, data on the usage of contraceptives in Ethiopia were collected using nationally representative surveys with participants.

previous research on the factors that influence the informed choice of contraceptive methods has shown that the following factors significantly influence the informed choice of contraceptive methods: husband's occupation, residency, types of contraceptive use, wealth index, media exposure, maternal education, source of contraceptive method, and visits to health facilities within the last year [18–21]. Diverse approaches have been used worldwide to boost the use of contraceptives; nonetheless, the incidence of these techniques in Sub-Saharan Africa (SSA) is still unacceptably low, and the false perception of their side effects is a serious public health issue [22,23]. As far as we are aware, there is very little information on the informed choice of contraceptive methods in the eastern African region, despite the fact that studies have been done on the utilization, unmet need, and discontinuation of contraceptive methods in SSA nations.

Therefore, the aim of this study was to ascertain the magnitude of informed choice of modern contraceptive methods user and its determinants among women aged 15–49 years who are currently using selected modern such as (Pill, injectable, IUD, Norplant/ implant, female sterilization) in Easter African countries.

## Method

### Patient and public involvement statement

In this study, the data were taken from Demographic and Health Surveys (DHS) data, which is secondary data.

### Study design, study area, and period

According to the most current demographic health study Multilevel mixed effect analysis was performed using data from the eight East African nations where research was done between 2012 and 2020. Every five years, the DHS collects community-based cross-sectional research data in order to generate updated health and health-related indicators.

### Data source, study population and sampling technique

The secondary data analysis was conducted using the DHS statistics for East African nations from 2012 to 2020. We made use of DHS surveys from eight nations in East Africa: Burundi, Comoros, Ethiopia, Malawi, Rwanda, Uganda, Zambia, and Zimbabwe. The data were appended to filter out the magnitude and determinant factors of among reproductive

age women who were informed choice among modern contraceptive usage in East African nations. The survey for every country contains different datasets. DHS deploys a stratified two-stage cluster design that includes enumeration areas as the first stage and generates a sample of households from each enumeration area as the second stage. The variable modern contraceptive (v312) from maternal recode (IR) data set was recoded to drop other than pill IUD, injectable, female sterilization, implant/Norplant contraceptive methods. A binary logistic regression model was used to identify determinants a factor to informed choice of modern contraceptive users. Determinants of informed choice were reported in terms of an AOR with a significance level of (95%). In the univariate analysis, at 95% confidence intervals with a p-value of < 0.25 was considered a candidate for the multivariable analysis of data. All variables with p values <0.05 were considered statistically significant. A total weighted sample of 6154 women was included in the study (Table 1).

### Study variable

**Outcome variables.** An informed choice of modern of contraception methods was the study's outcome. The six contemporary methods of contraception female sterilization, implant/Norplant, injectable contraceptive, IUD, pill, and emergency contraceptive were the subjects of a study on informed choice making among reproductive women aged 15–49. The DHS questionnaire was utilized to develop it, with responses obtained from the participants. Four factors were computed to define how the outcome variables should be measured (Table 2).

**Table 1. The weighted Sample size for magnitude and determinant factors of among reproductive age 15-49 years women with modern contraceptive user in Eastern Africa countries.**

| Country | years of survey | weighted sample (n) | weighted sample (%) |
|---|---|---|---|
| Burundi | 2016/2017 | 442 | 7.2 |
| Ethiopia | 2016 | 1454 | 23.7 |
| Comoros | 2012 | 99 | 1.61 |
| Malawi | 2015/2016 | 948 | 15.45 |
| Rwanda | 2019/2020 | 905 | 15 |
| Uganda | 2016 | 909 | 15 |
| Zambia | 2018 | 490 | 7.98 |
| Zimbabwe | 2015 | 887 | 14.46 |
| **Total** | | **6154** | **100%** |

**Table 2. Determination and computation Outcome Variables Used in the DHS datasets.**

| Variable | Explanation | Measurement |
|---|---|---|
| Informed about side effect | **when Started the current method,** 1.were you told about side effects (**v3a02**) | Yes = 1 (informed) No = 0 (not informed) |
| Informed by health provider | 2.Told about side effects by health or family planning worker (**v3a03**) | Yes = 1 (informed) No = 0 (not informed) |
| Informed how to deal with | **3.** Told how to deal with side effects (**v3a04**) | Yes = 1 (informed) No = 0 (not informed) |
| Informed other available method | **When you started the current method,** 4. were you told about other methods of family planning that you could use (**v3a05**) | Yes = 1 (informed) No = 0 (Not informed) |
| Informed other available method | 5. Were you ever told by a health or family-planning worker about other methods of family planning that you could use (**v3a06**) | Yes = 1 (informed) No = 0 (Not informed |
| Informed choice for modern contraception | by computation of five variable such **v3a02, v3a03**, **v3a04, v3a05, v3a06**. | Final outcome 1 = Yes 0 = No |

**Independent variable.** Hence DHS data are hierarchical, independent variables were individual and community levels considering for this analysis. The individual-level independent variables were reproductive women's age (15–19, 20–35, 36–49), Maternal educational status (No formal education, Primary education, Secondary and Higher education), Maternal occupation (House wife, employee, Others,), Religion (Cristian, Muslim, Others), Wealth index (Poor, Middle, Distance to health facility (Big problem, Not a big problem), media exposure (yes, no), intermate utilization (yes, no), Residency (urban, rural), Husband educational status (No formal education, Primary education, secondary and higher education) Occupational status (No working, Employed, house wife, Other), Husband occupational status (No working, Employed, Other), Contraceptive Decision (women, husband/partner, Joint decision), Currently Contraceptive (No, Yes) Visiting health facility in 12 month (No, Yes)

**The community-level variables.** Community poverty level (Low, High), Community illiteracy level (Low, High), Community media exposure, (Low, High), Community distance from health facility (Low, High), Place of residency (Urban, rural) in East African countries.

## Data processing and statically analysis

Stata version 14 statistical software was used to record, clean, and analyze the data that were taken from DHS data sets. The variables in the DHS data are arranged into clusters, with the similarity between variables inside a cluster being greater than that of variables outside of it. In order to do a conventional logistic regression analysis, the presumptions of independent data and uniform variance among clusters were violated. This suggests that taking in to account between-cluster effects requires the use of a complex model. In light of this, multilevel mixed-effects logistic regression was employed to identify the variables linked to knowledgeable selection of contemporary contraceptives.

Multilevel mixed effect logistic regression follows four models: the null model (only outcome variable), mode I (individual level variables), model II (community level variables), and model III (both individual and community level variables). The null model without independent variables was used to check the variability of informed choice of modern contraceptive method across the cluster. The relationship of individual-level variables with outcome variable (Model I) and the association of community-level variables with outcome variable (Model II) were determined. In the final model, the association of both individual and community-level variables was fitted with the outcome variable.

## Random effects

The intra-class correlation coefficient (ICC), proportional change in variance, and median odds ratio (MOR) were used to assess random effects or measures of variation of the outcome variables (PCV) To quantify the variation between clusters, the intra-cluster correlation coefficient (ICC) and proportionate change in variance (PCV) were calculated. When considering each cluster as a random variable, the ICC shows that the informed choice variance between clusters was computed as; $ICC = \frac{VC}{VC+3.29} \times 100$ % [24]. The MOR is the median value of the odds ratio between the area of the highest risk and the area of the lowest risk for informed choice of modern contraceptive when two clusters are randomly selected, using clusters as a random variable; $MOR = e^{0.95\sqrt{VC}}$ [24]. In addition to this, the PCV reveals the variation in the magnitude of informed choice of modern contraceptive usage explained by factors and computed as; $PCV = \frac{Vnull-VC}{Vnull} \times 00$ %; [24] where Vnull = variance of the null model and VC = cluster level variance [25–27]. The likelihood of making an informed decision and independent variables at the individual and community levels were estimated using the fixed effects. Adjusted odds ratio (AOR) and 95% confidence intervals with a p-value of less than 0.05 were used to evaluate and present the data. Deviance = -2 (log likelihood ratio) was used to compare the models due to the nested nature of the model, and the model with the lowest deviance was chosen as the best-fitted model. By evaluating the variance inflation factors (VIF), the multi-collinearity of the variables employed in the models was confirmed, and the results were found to be within an acceptable range of 1–10.

### Ethical approval and consent to participate

There was no need for ethical approval because this study was based on an analysis of survey datasets that were already in the public domain (EDHS) and readily accessible online with all identifying information anonymized.

### Result

This study included 6154 women of reproductive age who made an informed decision to use a modern type of contraception. Over two thirds (71.18%) of the women in the survey were from rural East Africa, and over half (51.9%) had only completed their primary education. Additionally, over 90% of participants did not use the internet, and over half (65.45) of the women had visited a health facility in the year prior to the survey (Table 3)

The magnitude of informed choice among reproductive age women who used modern contraceptive method such as pill, IUD, injectable, implant/Norplant in Eastern African countries found to be 20.70 (95% CI: (19.65, 21.78) (Fig 1)

### Random effect (Measures of variation) and model fitness

To determine whether the data supported the choice to evaluate randomness at the community level, a null model was employed. Significant variations in informed choice were found between communities according to the null model's results, which had a variance of.0883961 and a P value of 0.000. While the variance across clusters accounted for 2.61% of the overall cluster variation, the variance within clusters provided 11.56%. The probabilities of making an educated decision varied by a factor of times in the null model between higher and lower risk clusters. According to Model I's intraclass correlation value, the differences between communities are explained by 0.76% of the variation in informed choice. Next, we created Model II using community-level variables and the null model. The ICC value from Model II indicated that 16.61% of the differences in informed choice were due to cluster variability. The likelihood of informed choice varied 1.69 times between low and high informed choice in clusters in the final model (model III), which ascribed both individual and community-level variables to about 79.44% of the variation in the likelihood of informed choice (Table 4)

### Determinants of individual and community level factors with informed choice of among reproductive age women who used modern Contraceptive in East Africa countries

Maternal ages of 20–35, 36–49, secondary and higher education, media exposure, and visiting a health facility were found to be significantly associated with informed choice of modern contraceptive among reproductive age women in east African countries, according to the final fitted model of multivariable multilevel logistic regression.

The odds of informed choice were 2.022 times higher among reproductive women aged 20–35 years compared to women aged 15–19 years **(AOR** = 2.02 95% CI: 1.39–2.92). The odds of informed choice were 2.44 times more likely to occur among reproductive women age 15–19 years of age **(AOR** = 2.44, 95% CI, 1.66–3.61) and the odds of informed choice were 1.37 times higher who had secondary and higher education level compared to women who had no any education **(AOR** = 1.37 95% CI 1.02–1.84). A reproductive age woman who had media exposure were 1.25 times more likely to informed choice of modern contraceptive compared to a woman who had not media exposure **(AOR** = 1.25 95% CI: (1.05–1.49). and women who were visiting health facility 1.32 times more likely informed choice for modern contraceptive compared to women who were no visiting health facility within 12 months of study period **(AOR** = 1.32 95% CI (1.11–1.54) (Table 5**).**

### Discussion

Informed choice of modern contraceptive method contributes significant problem among reproductive age women in developing countries mainly in Eastern Africa. The aim of this research was to ascertain the magnitude and determinant factors modern contraceptive informed choice among reproductive age women in Eastern Africa countries. In this study,

**Table 3. Sociodemographic and economic characteristics of reproductive women who had informed choice modern contraceptive user in East Africa countries.**

| Individual and community level variables | frequency(n) | percentage (%) |
|---|---|---|
| **Age in year** | | |
| 15-19 | 453 | 8.08 |
| 20-35 | 3852 | 68.73 |
| 36-49 | 1299 | 23.17 |
| **Residency** | | |
| Urban | 1,615 | 28.81 |
| Rural | 3989 | 71.18 |
| **Religion** | | |
| Cristian | 3243 | 57.87 |
| Muslim | 709 | 12.65 |
| other | 1652 | 29.47 |
| **Educational Status** | | |
| No formal education | 1041 | 18.57 |
| Primary education | 2909 | 51.9 |
| Secondary and high | 1654 | 29.51 |
| **Husband educational status** | | |
| No formal education | 790 | 16.38 |
| Primary education | 2165 | 44.9 |
| secondary and higher | 1867 | 38.71 |
| **Occupational status** | | |
| No working | 1442 | 25.73 |
| Employed | 2091 | 37.31 |
| house wife | 81 | 1.44 |
| Other | 830 | 18.81 |
| **Husband occupational status** | | |
| No working | 353 | 6.29 |
| Employed | 2517 | 44.91 |
| Other | 2737 | 48.84 |
| **Wealth index** | | |
| Lower | 2101 | 37.49 |
| Middle | 1060 | 18.91 |
| Higher | 2443 | 43.59 |
| **Media exposure** | | |
| No | 2053 | 36.63 |
| Yes | 3551 | 63.36 |
| **Intermate utilization** | | |
| No | 5,009 | 90.99 |
| Yes | 496 | 9.00 |
| **Contraceptive Decision** | | |
| Female | 1009 | 20.94 |
| Male/partner | 343 | 7.12 |
| Joint decision | 3465 | 71.93 |
| **Currently Contraceptive** | | |
| No | 4263 | 78.42 |
| Yes | 1180 | 21.67 |

*(Continued)*

**Table 3.** (Continued)

| Individual and community level variables | frequency(n) | percentage (%) |
|---|---|---|
| **Visiting health facility 12 month** | | |
| No | 1,936 | 34.54 |
| Yes | 3668 | 65.45 |
| **Distance from health facility** | | |
| Big problem | 2113 | 37.70 |
| Not big problem | 3491 | 62.29 |
| **community poverty level** | | |
| Low | 2914 | 51.99 |
| High | 2690 | 48.00 |
| **Community illiteracy level** | | |
| Low | 2332 | 41.61 |
| High | 3227 | 57.58 |
| **Community media exposure** | | |
| Low | 2200 | 39.25 |
| High | 3404 | 60.74 |
| **Community distance health facility** | | |
| Low | 2716 | 48.46 |
| High | 2888 | 51.53 |

the magnitude of informed choice among reproductive age women found to be 20.70 (95% CI: (19.65, 21.78). This finding revealed that substantial number of reproductive age women using modern contraception method were uninformed choice about modern contraceptive methods, side effects and other concerned issue of family planning, consequently, discontinuation of service use [28]. This finding was lower than the previous studies conducted in sub-Saharan Africa 49.47% [10], multilevel study in Ethiopia 36.2% [18], 30% [29] UNFPA study in 24 countries 52% [30]. This discrepancy might be due to the deference in socio-economic and demographic condition, and variation in health infrastructure and health care service policy and the health service coverage, quality of maternal healthcare services, and economic and health policies of UNFPA the are better compared with those of Eastern Africa countries. On the other hand, this study was in line with study conducted southern Ethiopia, Sidama Region 23.5% [31]

In the final model of multilevel logistic regression analysis variables such as Maternal age, Maternal education, having media exposure, had visiting health facility within 12 months were found to be significantly associated with informed choice of modern contraceptive method user. Given this, the odds of having an informed choice of modern contraceptive among women 20–35 and 36–49 years old were 2.02 times (AOR = 2.02 CI: 1.39, 2.93) and 2.44 times (AOR = 2.44, 95% CI: 1.66, 3.61) higher as compared to 15–19 years old women respectively. This find was consistent with study done in Ethiopia [18], sub-Saharan countries [10], India [21]. As the women's age increases, the chance of having an information, experience and learning about contraceptive also increases also one study conclude that, healthcare providers marginalize attitudes toward younger women [21,32]. At the same time, women who attend secondary & higher education were 1.37 times (AOR = 1.37 95% CI 1.02, 1.84) higher odds of informed choice of modern contraceptive methods user compared to women who had no formal education. This is consistent, with studies done elsewhere [10,18,21,33]. The possible reason could be higher educated women more likely to engage with health care provider communication and understand information's they receive from their health care provider about informed choice of modern contraceptive, in addition, Educated women were better information access from other sources like media [19,34]. This result indicated that the importance of womens education for enhancing informed choice of modern contraceptive methods in East African countries.

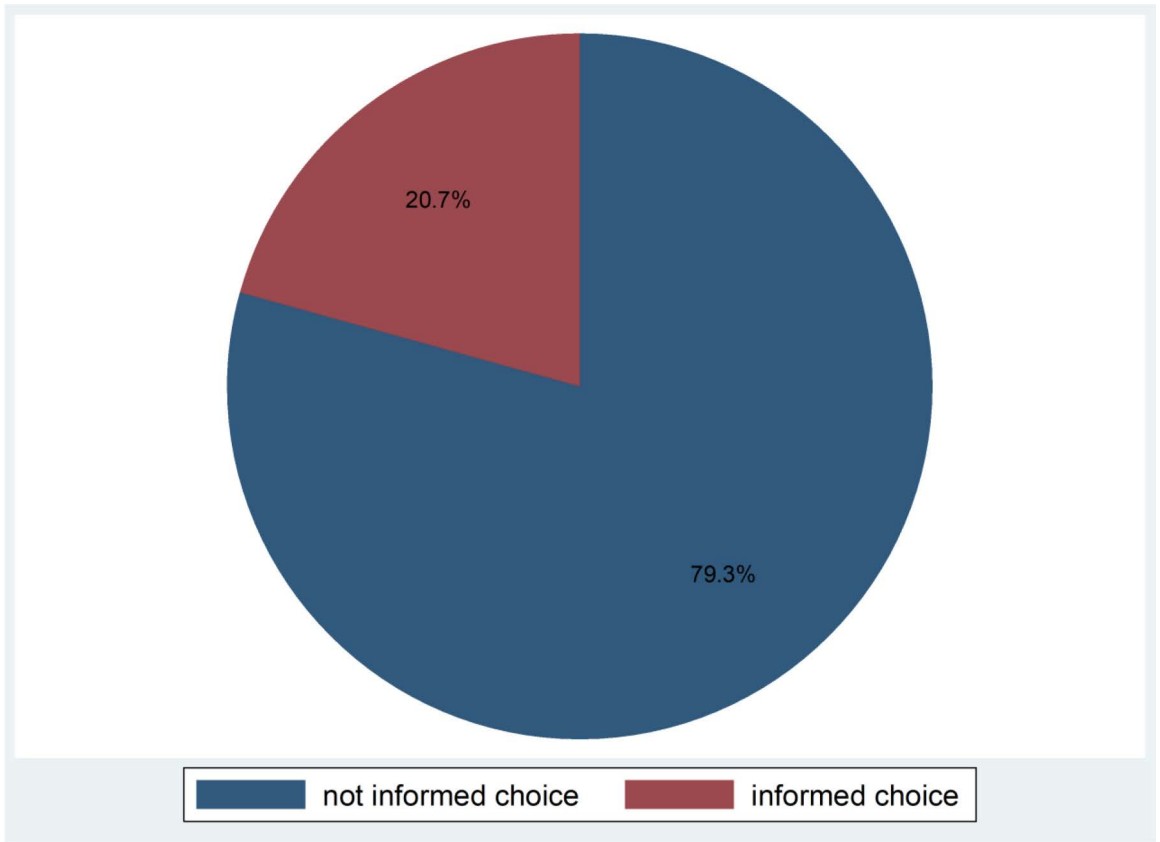

**Fig 1. Magnitudes of Informed choice of modern Contraceptive Methods among reproductive age women in Eastern Africa countries: A Multilevel Analysis of Demographic and Health Survey.**

**Table 4. Model comparison and random effect analysis for informed choice of modern contraceptive among reproductive age women in East African countries.**

| Parameter | Null model | model I | model II | model III |
|---|---|---|---|---|
| Variance | .0883961 | 0.053887 | 0.655594 | 0.430142 |
| ICC | 2.61% | 1.61% | 16.61% | 11.56% |
| MOR | 0.76 | 0.60 | 2.09 | 1.69 |
| PCV | Reference | 64.03% | 86.52% | 79.44 |
| LLR | -2855.5044 | -2353.2748 | -2839.6822 | -2349.9148 |
| Deviance | 5711.008 | 4706.5496 | 5679.3644 | 4699.8296 |

**ICC:** intra-cluster correlation, **MOR:** median odds ratio, **PCV:** proportional change in variance

**LLR:** loglikelihood ratio, **Deviance:** LLR *-2

Similarly, women who had media exposure were 1.25 times (AOR = 1.25 95% CI 1.05, 1.49) higher odds of informed choice of modern contraceptive methods user as compared to women who had no media exposure. This finding was similar with study done in Ethiopia (EDHS 2016) and sub Saharan Africa [10,35]. The possible justification for thus association was, women who had media exposure could be easily retained information about different types of contraceptive methods and their side effects, which enabled them to make an informed of contraceptive methods [36]. In addition to this, women

**Table 5. Multivariable multilevel logistic regression analysis of individual-level and community level variable associated with informed choice among reproductive women who use modern contraceptive in East African countries.**

| Individual and community level factors | Model I AOR (95% CI) | Model II AOR (95% CI) | Model III AOR (95% CI) |
|---|---|---|---|
| **Maternal age** | | | |
| 15-19 | 1 | | **1** |
| 20-35 | 2.01(1.39, 2.91) | | **2.02 (1.39, 2.93) **** |
| 36-49 | 2.45 (1.66, 3.62) | | **2.44 (1.66, 3.61) **** |
| **Residency** | | | |
| Urban | | 0.79 (0.69, 0.92) | 1.13 (0.92-1.40) |
| Rural | | 1 | 1 |
| **Maternal educational level** | | | |
| No education | 1 | | 1 |
| primary education | 1.22 (0.97, 1.54) | | 1.19 (0.94, 1.5) |
| secondary and higher | 0.20 (1.06, 1.89) | | **1.37 (1.02, 1.84) **** |
| **Husband education** | | | |
| No education | 1 | | 1 |
| Primary education | 0.89 (0.77, 1.25) | | 0.98(0.77, 1.25) |
| Secondary & higher | 0.15 (0.83, 1.45) | | 1.08(0.82, 1.42) |
| **Marital occupation** | | | |
| No working | 1 | | 1 |
| employed | 1.05 (0.99, 1.25) | | 1.04(0.87, 1.24) |
| house wife | 0.71 (0.33, 1.50) | | 0.7(0.33, 1.49) |
| Other | 1.10 (0.89, 1.37) | | 1.08 (0.86, 1.34) |
| **Husband occupation** | | | |
| no working | 1 | | 1 |
| employed | 0.78 (0.45, 1.50) | | 1.14 (0.84, 1.55) |
| other | 0.74 (0.43, 1.78) | | 1.15 (0.84, 1.57) |
| **Wealth index** | | | |
| lower | 1 | | 1 |
| middle | 1.04 (0.84, 1.28) | | 1.06 (0.86, 1.31) |
| rich | 0.93 (0.76, 1.15) | | 0.95 (0.77, 1.18) |
| **Media Exposure** | | | |
| No | 1 | | 1 |
| Yes | 1.28 (1.08, 1.53) | | **1.25 (1.05, 1.49**) ** |
| **Internet use** | | | |
| No | 1 | | 1 |
| Yes | 1.11(0.85, 1.47) | | 1.14 (0.84, 1.46) |
| **Decision maker on Contraceptive** | | | |
| women | 1 | | 1 |
| Husband/partner | 1.26(0.91, 1.72) | | 1.26 (0.92, 1.73) |
| Joint decision | 1.07 (0.89, 1.29) | | 1.08 (0.90, 1.30) |
| **Health facility visiting status** | | | |
| No | 1 | | 1 |
| Yes | 1.32 (1.12, 1.55) | | **1.31 (1.11, 1.54) **** |
| **Distance from health facility** | | | |
| Big problem | 1 | | 1 |
| Not big problem | 0.90 (0.77, 1.06) | | 0.90 (0.76, 1.75) |

*(Continued)*

**Table 5.** (Continued)

| Individual and community level factors | Model I AOR (95% CI) | Model II AOR (95% CI) | Model III AOR (95% CI) |
|---|---|---|---|
| **Community level variable** | | | |
| **Community poverty level** | | | |
| Low | | 1 | 1 |
| High | | 1.17 (1.00, 1.37) | 1.15 (0.97, 1.37) |
| **Community illiteracy level** | | | |
| low | | 1 | 1 |
| High | | 0.78 (0.68, 0.91) | 0.87 (0.74, 1.02) |
| **Community Media Exposure** | | | |
| Low | | 1 | 1 |
| High | | 0.88 (0.75, 1.03) | 0.88 (0.74, 1.04) |
| **Community distance health facility** | | | |
| Low | | 1 | 1 |
| High | | 0.98 (0.85, 1.14) | 1.03 (0.88, 1.21) |
| **East African countries** | | | |
| Low income | | 1 | 1 |
| lower middle-income countries | | 1.11 (0.96, 1.27) | 1.03 (0.87, 1.21) |

who visit health facility within 12 months were 1.31 times (AOR = 1.31 95% CI: 1.11, 1.54) higher odds of informed choice of modern contraceptive methods user as compared to those who were not visiting health facility within 12 months of survey. This finding was consistent with study in Ethiopia (EDHS 2016). The justification of this finding could be, women's who had trained visiting health facility could easily obtain or access more information about different types of contraceptive methods, their preferable choice about contraception [36].

## Strength and limitation

Data from national health surveys and demographic were used in this investigation. As with any cross-sectional study, there are limitations. First, the findings on informed choice only include a subset of modern contraceptive methods, such as pills, IUDs, implants, implants/Norplant, injectables like Depo-Provera, and female sterilization. Moreover, recall bias was not avoided and a causal relationship was not determined.

## Conclusion

Many East African women between the ages of 15 and 49 who are fertile were not well informed about the harmful effects of modern contraceptives methods or other options. Only 20.7% of the women in this survey had sufficient knowledge about the negative effects of modern contraceptives methods and the options that healthcare practitioners may supply. The following factors were found to be significantly associated with an informed choice of modern contraceptive methods; such as maternal age, maternal educational status, media exposure, and visiting a health facility within a year.

## Recommendation

Health care facilities should prioritize empowering women to make an informed choice in order to ensure that the service is continued. In order to assist women in making informed decisions about contraceptive options, providers will also use counseling sessions.

## Supporting information

**S1 Fig. Mini data (Data used for analysis).**
(XLS)

## Author contributions

**Conceptualization:** Gebreeyesus Abera Zeleke, Birtukan Atena Negash, Berhan Tekeba, Tadesse Tarik Tamir, Workie Zemene Worku, Alebachew Ferede Zegeye.

**Data curation:** Gebreeyesus Abera Zeleke, Berhan Tekeba, Workie Zemene Worku, Alebachew Ferede Zegeye.

**Formal analysis:** Gebreeyesus Abera Zeleke, Birtukan Atena Negash, Kassaw Belay Shiferaw, Berhan Tekeba, Workie Zemene Worku, Alebachew Ferede Zegeye.

**Investigation:** Gebreeyesus Abera Zeleke, Workie Zemene Worku, Alebachew Ferede Zegeye.

**Methodology:** Gebreeyesus Abera Zeleke, Kassaw Belay Shiferaw, Workie Zemene Worku, Alebachew Ferede Zegeye.

**Software:** Gebreeyesus Abera Zeleke, Berhan Tekeba, Tadesse Tarik Tamir, Workie Zemene Worku, Alebachew Ferede Zegeye.

**Supervision:** Gebreeyesus Abera Zeleke, Berhan Tekeba, Workie Zemene Worku, Alebachew Ferede Zegeye.

**Visualization:** Gebreeyesus Abera Zeleke, Tadesse Tarik Tamir, Workie Zemene Worku, Alebachew Ferede Zegeye.

**Writing – original draft:** Gebreeyesus Abera Zeleke, Birtukan Atena Negash, Tadesse Tarik Tamir, Alebachew Ferede Zegeye.

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
