## [Decision Letter · Decision Letter 0]

8 Dec 2024

PONE-D-24-06637Informed choice of modern Contraceptive Methods and determinant factors among reproductive age women in Eastern Africa countries: A Multilevel Analysis of Demographic and Health Survey.PLOS ONE

Dear Dr. zeleke,

Thank you for submitting your manuscript to PLOS ONE. After careful consideration, we feel that it has merit but does not fully meet PLOS ONE’s publication criteria as it currently stands. Therefore, we invite you to submit a revised version of the manuscript that addresses the points raised during the review process.

We look forward to receiving your revised manuscript.

Kind regards,

Yitagesu Habtu Aweke, Ph.D

Academic Editor

PLOS ONE

6. We note you have included a table to which you do not refer in the text of your manuscript. Please ensure that you refer to Table 5 in your text; if accepted, production will need this reference to link the reader to the Table.

Additional Editor Comments:

You reported an “informed choice” prevalence of 20.70% from DHS data spanning 2012 to 2020 across eight East African countries. Meanwhile, a study covering nine East African countries (DHS 2015–2021, DOI: 10.1371/journal.pone.0297018) reported an overall modern contraceptive prevalence of 45.68%. How do you explain such a significant gap between informed choice and modern contraceptive use, given that family planning counselling is expected to ensure informed decision-making at initiation?Your study excluded Madagascar and Tanzania but included Comoros. What was the rationale behind selecting these countries? Was it related to data availability, methodological concerns, or another reason?You used DHS data from 2012 to 2020 despite the availability of more recent data (2015–2021).Why did you choose an earlier timeframe instead of leveraging the latest data?Could this choice affect the relevance of your findings? Please elaborate further on how your study adds new insights to the existing body of knowledge on family planning and informed choice. What gaps does it fill over the existing data?Could you explain more clearly how you defined and operationalised your outcome variables? This clarification will help in understanding the study's methodology and its implications.

Reviewers' comments:

Reviewer's Responses to Questions

**Comments to the Author**

1. Is the manuscript technically sound, and do the data support the conclusions?

Reviewer #1: Yes

Reviewer #2: Yes

2. Has the statistical analysis been performed appropriately and rigorously? 

Reviewer #1: Yes

Reviewer #2: I Don't Know

3. Have the authors made all data underlying the findings in their manuscript fully available?

Reviewer #1: Yes

Reviewer #2: Yes

4. Is the manuscript presented in an intelligible fashion and written in standard English?

Reviewer #1: Yes

Reviewer #2: No

5. Review Comments to the Author

Reviewer #1: Dear author

Conducting a study on contraceptive methods is required in all countries. Therefore, conducting this study is commendable. Below are comments to improve the article.

Abstract:

1. Please correct this sentence:

This study concluded that the proportion of reproductive age women using modern contraceptive techniques made an informed choice in eight nations in Eastern Africa.

Introduction

1. Do you have statistics on unwanted pregnancies in East Africa? If you have statistics, please write.

2. Do you have statistics on illegal abortions? If you have statistics, please write.

3. It seems that illegal abortions following uninformed choice of contraceptive methods are your concern and this is the necessity of your study. Please explain more about this issue. Explain that illegal abortions threaten women's reproductive health.

Methods

1. Barrier methods and vasectomy methods are not among the methods?

Discussion

1. Why is there no discussion of non-significant variables?

Reviewer #2: � Comments to Authors:

Abstract:

• please correct the linguistic error of repetition in the following phrase “Data from Data from the recent DHS” stated in the subtitle “method”.

Title:

• The title of the study is clear, inclusive and precise to the study’s objectives and aims. The authors stated the study design distinctly. However, concerns are regarding using “modern” description for a well-known methods of contraception.

Introduction:

• The rationale and aim of the study are well stated.

• The introduction would benefit from a more comprehensive description of how family planning services are being delivered in the research area; this would help make the context clearer for readers and researchers unfamiliar with Sub-Saharan Africa.

• While the idea of informed choice when choosing contraception method has been stated several times as an important factor, further elaboration is needed on its broader effects, beyond just preventing unwanted pregnancies. Specifically, a more direct link between informed choice and lower contraceptive discontinuation rates would strengthen the argument.

• The inclusion of statistical data from countries like India, Bangladesh, Kenya, Ethiopia, and sub-Saharan Africa adds strength to the argument. However, there is a lack of explanation about these statistics. How were these numbers gathered? Clarifying how informed choice was ensured and its relationship to better contraceptive choices would add credibility and impact to the discussion.

Methods:

• The methods section provides a comprehensive description of the study design, data source, population, and statistical analysis. Moreover, Tables summarizing key variables are quiet helpful.

• The authors clearly explain the weighted sample size, which helps strengthen the reliability of the method.

• The explanation of the statistical models (e.g., multilevel mixed-effects logistic regression) is detailed but might be difficult to follow for readers unfamiliar with advanced statistical methods.

• Also, the use of specialized terms such as "proportional change in variance" (PCV), "intra-class correlation coefficient" (ICC), and "variance inflation factors" (VIF) may be challenging for some readers. It would be beneficial to define these terms more clearly, when they first appear or provide references for readers unfamiliar with them.

• Regarding “outcome” variables, the authors conducted a questionnaire survey that would define and assess “informed choice” such as; knowing the side effects and other available methods. Nevertheless, many other aspects are worth mentioning in order to label it as “informed choice”. These aspects includes but not limited to:

Access to information: providing clear details about each method, each method , its effectiveness, side effects, and any medical or personal considerations.

Understanding Options: Ensuring that individuals understand the information provided and can ask questions to clarify any uncertainties.

Freedom of choice: must have the autonomy to choose the method that best suits their personal needs and preferences without any pressure from healthcare provider or a family member.

• While the methodology is well-detailed, the section could benefit from a brief mention of potential limitations in the way informed choice is measured.

• Overall, the methods section is well-organized and provides a thorough description of the study's design and analysis. With a few clarifications and improvements, it would be more accessible to a wider audience.

Results:

• The authors conducted a thorough analysis of the results, supported by sufficient amount of data justifying the conclusion.

• The data are clearly presented by the authors, and the use of percentages for each variable enhances the clarity of the findings. The study's main result, showing the magnitude of informed choice among women using modern contraceptives, is well articulated (20.70%).

Discussion:

• The author effectively highlights key findings and discusses the demographic and socio-economic factors that contribute to this low rate.

• The comparison with other studies in sub-Saharan Africa, such as those in Ethiopia, India, and broader African contexts, is useful in contextualizing the results and offering possible explanations for the discrepancies observed.

• While the authors mentioned multiple contributing socio-economic factors, it would be beneficial to delve deeper into how specific health policies or community-level interventions could address the lack of informed choice, particularly in rural areas. This could provide actionable insights for policymakers.

• Overall, the discussion section is informative and offers a solid interpretation of the study results.

Strength and Limitation:

• The acknowledgment of limitations such as the potential for recall bias demonstrates transparency and a clear understanding of the study's constraints. This might provide a guidance for future researches.

Conclusion:

• Correctly answered the research question.

• The authors effectively highlights the factors associated with informed choice, such as maternal age, education, media exposure, and health facility visits.

• Including recommendations and suggestions for improving informed choice aimed at healthcare policymakers and media representatives would enhance the conclusion.

Minor:

• Some minor grammatical issues could be addressed to improve readability.

• English editing and proofreading by a native English speaker needed. The authors should correct some typos and language mistakes.

6. PLOS authors have the option to publish the peer review history of their article (what does this mean? ). If published, this will include your full peer review and any attached files.

**Do you want your identity to be public for this peer review?** For information about this choice, including consent withdrawal, please see our Privacy Policy .

Reviewer #1: No

Reviewer #2: No

---

## [Decision Letter · Decision Letter 1]

12 Mar 2025

PONE-D-24-06637R1Informed choice of modern Contraceptive Methods and determinant factors among reproductive age women in Eastern Africa countries: A Multilevel Analysis of Demographic and Health Survey.PLOS ONE

Dear Dr. zeleke,

Thank you for submitting your manuscript to PLOS ONE. After careful consideration, we feel that it has merit but does not fully meet PLOS ONE’s publication criteria as it currently stands. Therefore, we invite you to submit a revised version of the manuscript that addresses the points raised during the review process.

We look forward to receiving your revised manuscript.

Kind regards,

Yitagesu Habtu Aweke, Ph.D

Academic Editor

PLOS ONE

Journal Requirements:

Reviewers' comments:

Reviewer's Responses to Questions

**Comments to the Author**

1. If the authors have adequately addressed your comments raised in a previous round of review and you feel that this manuscript is now acceptable for publication, you may indicate that here to bypass the “Comments to the Author” section, enter your conflict of interest statement in the “Confidential to Editor” section, and submit your "Accept" recommendation.

Reviewer #2: All comments have been addressed

2. Is the manuscript technically sound, and do the data support the conclusions?

Reviewer #2: (No Response)

3. Has the statistical analysis been performed appropriately and rigorously? 

Reviewer #2: (No Response)

4. Have the authors made all data underlying the findings in their manuscript fully available?

Reviewer #2: (No Response)

5. Is the manuscript presented in an intelligible fashion and written in standard English?

Reviewer #2: (No Response)

6. Review Comments to the Author

Reviewer #2: (No Response)

7. PLOS authors have the option to publish the peer review history of their article (what does this mean? ). If published, this will include your full peer review and any attached files.

**Do you want your identity to be public for this peer review?** For information about this choice, including consent withdrawal, please see our Privacy Policy .

Reviewer #2: **Yes: ** Bayan Al Omari

---

## [Editor Report · Decision Letter 2]

16 Apr 2025

Informed choice of modern Contraceptive Methods and determinant factors among reproductive age women in Eastern Africa countries: A Multilevel Analysis of Demographic and Health Survey.

PONE-D-24-06637R2

Dear Dr. Gebreeyesus Abera,

We’re pleased to inform you that your manuscript has been judged scientifically suitable for publication and will be formally accepted for publication once it meets all outstanding technical requirements.

Kind regards,

Yitagesu Habtu Aweke, Ph.D

Academic Editor

PLOS ONE

Additional Editor Comments (optional):

Please make your recommendation under the subsection of "Conclusion".
---

## [Editor Report · Acceptance letter]

PONE-D-24-06637R2

PLOS ONE

Dear Dr. Zeleke,

I'm pleased to inform you that your manuscript has been deemed suitable for publication in PLOS ONE. Congratulations! Your manuscript is now being handed over to our production team.

Kind regards,

on behalf of

PhD Candidate Yitagesu Habtu Aweke

Academic Editor

PLOS ONE